# Low p16 Cytoplasmic Staining Predicts Poor Treatment Outcome in Patients with p16-Negative Locally Advanced Head and Neck Squamous Cell Carcinoma Receiving TPF Induction Chemotherapy

**DOI:** 10.3390/biomedicines11020339

**Published:** 2023-01-25

**Authors:** Yen-Hao Chen, Chih-Yen Chien, Tai-Ling Huang, Tai-Jen Chiu, Yu-Ming Wang, Fu-Min Fang, Shau-Hsuan Li

**Affiliations:** 1Division of Hematology-Oncology, Department of Internal Medicine, Kaohsiung Chang Gung Memorial Hospital, College of Medicine, Chang Gung University, Kaohsiung 833, Taiwan; 2School of Medicine, College of Medicine, Chang Gung University, Taoyuan 333, Taiwan; 3School of Medicine, Chung Shan Medical University, Taichung 402, Taiwan; 4Department of Nursing, School of Nursing, Fooyin University, Kaohsiung 831, Taiwan; 5Department of Otolaryngology, Kaohsiung Chang Gung Memorial Hospital, College of Medicine, Chang Gung University, Kaohsiung 833, Taiwan; 6Department of Radiation Oncology, Kaohsiung Chang Gung Memorial Hospital, College of Medicine, Chang Gung University, Kaohsiung 833, Taiwan

**Keywords:** p16, head and neck cancer, squamous cell carcinoma, induction chemotherapy, TPF

## Abstract

Human papillomavirus (HPV) has been proven to be associated with head and neck squamous cell carcinoma (HNSCC), and diffuse p16 unclear staining is usually considered as HPV-positive. The aim of the current study was to investigate the role of p16 cytoplasmic staining in HNSCC prognosis. A total of 195 HNSCC patients who received docetaxel, cisplatin, and 5-fluouracil (TPF) induction chemotherapy followed by chemoradiotherapy were enrolled. The status of p16 cytoplasmic staining was determined using immunohistochemistry. The median follow-up was 26.0 months for the whole study population and 90.3 months for 51 living survivors. p16 cytoplasmic staining was low in 108 patients and high in 87 patients. Low expression of p16 cytoplasmic staining and primary tumor location in the oral cavity were both independent factors indicating a worse response rate to TPF induction chemotherapy in the univariate and multivariate analyses. The logistic regression model also showed that low expression of p16 cytoplasmic staining and clinical N2–3 status were independent prognostic factors for worse progression-free survival and overall survival. Our study showed that p16 cytoplasmic staining could predict the treatment response to TPF induction chemotherapy and is an independent prognostic factor of survival in HNSCC.

## 1. Introduction

Head and neck squamous cell carcinoma (HNSCC) is an aggressive, life-threatening malignancy worldwide and is the sixth leading cause of death in Taiwan [1]. The location of HNSCC includes the nasal cavity, oral cavity, oropharynx, hypopharynx, and larynx. Tobacco smoking, alcohol consumption, and betel nut chewing are well-known risk factors for HNSCC; however, not all causes of HNSCC are related to these behaviors. Human papillomavirus (HPV) is a nonenveloped double-stranded DNA virus that infects and replicates in skin and mucous membranes. The carcinogenicity of HPV was recognized as being associated with cervical cancer by Professor Haral zur Hausen in the 1970s. HPV infection was also identified in 1983 in patients with oral squamous cell carcinoma [2]. To date, more than 220 HPV types have been classified. Recently, growing evidence has confirmed an association between HPV and HNSCC, and the incidence of HPV-positive HNSCC has significantly increased, especially in the oropharynx [3,4,5].

p16, also called cyclin-dependent kinase inhibitor 2A (CDKN2A), is a cyclin-dependent kinase (CDK) inhibitor that arrests the cell cycle in the G1 stage. Its gene product inhibits CDK4/6 binding with cyclin D, resulting in phosphorylation of retinoblastoma (Rb). The dephosphorylated Rb subsequently entraps nuclear transcription factor E2F and blocks the ability to bind to other transcription factors to inhibit G1/S phase progression and prevent cell proliferation. In HPV+ HNSCC, the loss of viral regulatory E2 gene contributes to increased E7 oncoprotein, which binds phosphorylated Rb, and results in its dissociation from E2F [6]. The latter, in turn, upregulates the transcription of cell-growth- and proliferation-related genes, including cyclin A, cyclin E, and p53 [7]. This dysfunction of Rb results in a compensatory overexpression of nuclear p16 that is characteristic of HPV+ HNSCC [8].

Evidence has demonstrated that HPV + HNSCC is distinctly different from HPV-HNSCC [9,10,11,12]. First, less heavy tobacco use or alcohol consumption is mentioned in patients with HPV+ HNSCC compared with those with HPV-HNSCC; this finding also explains why the incidence of alcohol- or cigarette-related HNSCC declines but the number of HPV + HNSCC increases [13,14]. Second, patients with HPV + HNSCC are younger than those with HPV-HNSCC [15]. Third, the tumor location in these two groups is different: HPV + HNSCC is more common in the oropharynx, but other risk factors (tobacco- and alcohol-related HNSCC) more frequently arise in the oral cavity, hypopharynx, and larynx. Fourth, HPV + HNSCC has a better prognosis than HPV-HNSCC, including lower mortality and a decreased risk of relapse after treatment. A biomarker analysis study that involved a phase III trial (E1395) and a phase II trial (E3301) demonstrated that better progression-free survival (PFS) and overall survival (OS) were observed in HPV + HNSCC than in HPV-HNSCC, suggesting that HPV is a favorable prognostic factor in recurrent or metastatic HNSCC [16]. An Eastern Cooperative Oncology Group (ECOG) phase II trial showed that patients with HPV + HNSCC had higher response rates (RRs) to induction chemotherapy and chemoradiation therapy than those with HPV-HNSCC. Additionally, the HPV + HNSCC group had an improved OS and lower risk of disease progression than the HPV-HNSCC group, indicating that tumor HPV status is strongly associated with therapeutic response and survival [9]. Another phase II trial (E1308) that focused on induction chemotherapy followed by reduced-dose radiation and weekly cetuximab in HPV+ oropharyngeal cancer patients revealed that reduced-dose radiotherapy with concurrent cetuximab is reasonable in favorable-risk HPV + HNSCC patients who responded to induction chemotherapy, including better PFS, OS, and lower incidence of difficulty swallowing solids or impaired nutrition [17].

Currently, HPV detection is regarded as a standard test to determine the tumor stage of oropharyngeal cancer in the eighth edition of the American Joint Committee on Cancer (AJCC) [18]. HPV infection status is defined as positive or negative nuclear staining of p16 by immunohistochemistry; however, the role of cytoplasmic staining of p16 in the clinical outcome of HNSCC remains unclear. The present study aimed to investigate the role of p16 cytoplasmic staining in HNSCC prognosis.

## 2. Materials and Methods

### 2.1. Patient Population 

Patients with HNSCC who underwent TPF induction chemotherapy at Kaohsiung Chang Gung Memorial Hospital between January 2010 and December 2015 were retrospectively reviewed. Only squamous cell carcinoma in pathology and tumors located in the oral cavity, oropharynx, hypopharynx, and larynx were included; those in nasal cavity or of unknown primary origin were excluded. Additionally, patients with a history of a second primary cancer, whether before or after the diagnosis of HNSCC, were excluded. Patients were required to receive TPF induction chemotherapy rather than other chemotherapy regimens, and those with distant metastasis or undergoing other anticancer treatments were not enrolled. Finally, 195 patients with HNSCC who met the inclusion criteria were identified.

### 2.2. TPF Induction Chemotherapy and Chemoradiotherapy 

The TPF induction chemotherapy regimen was as follows: docetaxel 60 mg/m^2^ intravenous infusion on day 1, cisplatin 60 mg/m^2^ intravenous infusion on day 1, and 5-fluorouracil 600 mg/m^2^ 24 h intravenous infusion on days 1 to 4 every three weeks for three cycles, except in cases of intolerance to adverse events, disease progression, or withdrawal of consent by patients. Chemotherapy was administered according to a previously described protocol [19,20,21].

Chemoradiotherapy (CRT) was followed by TPF induction chemotherapy in each patient. The planned radiotherapy dose for the primary tumor was 70 Gy in 35 fractions (2 Gy, 5 days per week). The doses administered to the involved lymph nodes (LNs) and uninvolved LNs were 60–70 Gy and 50 Gy, respectively.

### 2.3. Immunohistochemistry 

Immunohistochemical staining was performed using immunoperoxidase. Staining was performed on slides (4 mm) of formalin-fixed, paraffin-embedded tissue sections using primary antibodies against p16, INK4 (BD Biosciences, San Jose, CA, USA). Briefly, after deparaffinization and rehydration, antigen retrieval was performed by treating the slides with 10 mM citrate buffer (pH 6.0) in a hot water bath (95 °C) for 20 min. Endogenous peroxidase activity was blocked for 15 min with 0.3% hydrogen peroxide. After blocking with 1% goat serum for 1 h at room temperature, sections were incubated overnight for at least 18 h with primary antibodies at 48 °C. Immunodetection was performed using an LSAB2 kit (Dako, Carpinteria, CA, USA), followed by reaction with 3-3’-diaminobenzidine for color development, and hematoxylin was used for counterstaining. Cervical carcinoma was used as a positive control, and esophageal cancer was used as a negative control. Assessment of staining was independently performed by two pathologists who were blinded to information regarding the clinicopathological features or prognosis of the patients. p16-negative (HPV-negative) expression was defined as <50% nuclear staining of tumor cells [22,23]. After excluding seven patients with p16-positive (HPV-positive) HNSCC, a semiquantitative immunoreactive score (IRS) was used to evaluate p16 cytoplasmic staining in 195 patients with p16-negative (HPV-negative) locally advanced HNSCC [24]. The IRS was calculated by multiplying the staining intensity (graded as: 0 = no staining, 1 = weak staining, 2 = moderate staining, and 3 = strong staining) and the percentage of positively stained cells (0 = no stained cells, 1 = ≤10% stained cells, 2 = 10–50% stained cells, 3 = 51–80% stained cells, and 4 = ≥80% stained cells). The criterion for high p16 cytoplasmic staining was an IRS score ≥6. 

### 2.4. Ethics Statement

This retrospective study was approved by the Chang Gung Medical Foundation Institutional Review Board (201900561B0). All procedures were performed in accordance with the ethical standards of the Institutional Research Committee and World Medical Association Declaration of Helsinki. The requirement for written informed consent was waived by the review board due to the retrospective design of this study.

### 2.5. Statistical Analysis

All statistical analyses in the current study were performed using SPSS software version 26 (International Business Machines Corp., Armonk, NY, USA). Differences in categorical variables were examined using the chi-square test. The Kaplan–Meier method was used for survival analysis, and the log-rank test was performed to test the differences. A Cox proportional hazards model was used to determine independent prognostic factors in the multivariate analysis. Odds ratios (ORs) with 95% confidence intervals (CIs) were calculated to test the strength of the association between the prognostic parameters and survival. PFS was defined as the duration between the start of TPF induction chemotherapy and the date of tumor recurrence or death from any cause, without evidence of recurrence. OS was calculated from the date of HNSCC diagnosis to death or the time of last living contact. All tests were two-sided, and statistical significance was set at *p* < 0.05.

## 3. Results

### 3.1. Patient 

Between January 2010 and December 2015, 195 HNSCC patients received TPF induction chemotherapy followed by CRT at Kaohsiung Chang Gung Memorial Hospital. All the patients had an Eastern Cooperative Oncology Group performance status score of zero or one. The median age was 52 years, and the primary tumor sites included oral cavity, oropharynx, hypopharynx, and larynx. Most patients were clinical T4a–T4b, and more than 80% of the whole population were stage IVA or IVB. The exposure to carcinogens included smoking, alcohol consumption, and betel nut chewing. Most patients benefited from TPF induction chemotherapy, including complete response (CR), partial response (PR), and stable disease (SD); progressive disease (PD) was found in few patients. The median follow-up duration was 26.0 months for the whole study population and 90.3 months for 51 living survivors. The characteristics of the 195 patients are shown in Table 1.

### 3.2. p16 Cytoplasmic Staining and Therapeutic Response 

The results of immunohistochemical staining for cytoplasmic p16 are shown in Figure 1. p16 cytoplasmic staining was low in 108 patients (55%) and high in 87 patients (45%). There were no significant differences between the two groups, including age, sex, clinical T status, clinical N status, tumor stage, primary tumor site, smoking, alcohol consumption, or betel nut chewing. A comparison of the results is presented in Table 2.

In the analysis of the response to TPF induction chemotherapy, the responder group (CR or PR) and non-responder group (SD or PD) were compared for baseline characteristics: no significant differences were noted for age, sex, clinical T status, clinical N status, tumor stage, smoking, alcohol, or betel nut chewing. Patients with low expression of p16 cytoplasmic staining had a worse response rate to TPF induction chemotherapy than those with high expression of p16 cytoplasmic staining (53% vs. 70%, *p* = 0.014). In addition, a lower response rate to TPF induction chemotherapy was noted in patients with primary tumors located in the oral cavity than in those with primary tumors arising from areas other than the oral cavity (52% vs. 67%, *p* = 0.043). Moreover, low expression of p16 cytoplasmic staining (OR: 2.12, 95% CI: 1.15–3.89, *p* = 0.016) and primary tumors located in the oral cavity (OR: 2.07, 95% CI: 1.13–3.82, *p* = 0.019) were both independent factors associated with a worse response rate to TPF induction chemotherapy in the multivariate logistic regression model. The results of univariate and multivariate analyses of response to TPF induction chemotherapy are shown in Table 3 and Table 4, respectively.

### 3.3. p16 Cytoplasmic Staining and Clinical Outcome

In the univariate analysis of PFS, age, sex, clinical T status, tumor stage, smoking, alcohol consumption, and betel nut chewing did not show statistical significance. Patients with low p16 cytoplasmic staining had a worse 5-year PFS rate than those with high p16 cytoplasmic staining (24% vs. 38%, *p* = 0.009, Figure 2A). A shorter 5-year PFS rate was found in patients with clinical N2–3 than in those with N0–1 (25% vs. 42%, *p* = 0.009). Patients with tumors arising from areas other than the hypopharynx/larynx had an inferior 5-year PFS rate than those with tumors located in the hypopharynx/larynx (26% vs. 41%, *p* = 0.045). Multivariate analysis showed that clinical N2–3 (OR: 1.64, 95% CI: 1.14–2.36, *p* = 0.007) and low expression of p16 cytoplasmic staining (OR: 1.47, 95% CI: 1.05–2.05, *p* = 0.024) were independent prognostic factors for a worse 5-year PFS rate.

With respect to OS, univariate analysis showed that there were no significant differences in age, sex, clinical T status, tumor stage, smoking, alcohol consumption, and betel nut chewing. An inferior 5-year OS rate was observed in patients with low p16 cytoplasmic staining than in those with high p16 cytoplasmic staining (26% vs. 45%, *p* = 0.009, Figure 2B). Patients with clinical N2–3 had a shorter 5-year OS rate than those with N0–1 (27% vs. 49%, *p* = 0.004). A worse 5-year OS rate was found in patients with tumors arising from areas other than the hypopharynx/larynx, than in those with tumors located in the hypopharynx/larynx (30% vs. 47%, *p* = 0.046). In the multivariate analysis, clinical N2–3 (OR: 1.75, 95% CI: 1.21–2.53, *p* = 0.003) and low expression of p16 cytoplasmic staining (OR: 1.49, 95% CI: 1.06–2.08, *p* = 0.021) were independent prognostic factors for a worse 5-year OS rate. The survival outcomes of the univariate and multivariate analyses are presented in Table 5 and Table 6, respectively.

According to the significance of p16 cytoplasmic staining and clinical N status in the multivariate Cox regression analysis for PFS and OS, these patients were divided into four groups: group 1 (high p16 expression + clinical N0–1, *n* = 32), group 2 (low p16 expression + clinical N0–1, *n* = 33), group 3 (high p16 expression + clinical N2–3, *n* = 55), and group 4 (low p16 expression + clinical N2–3, *n* = 75). Group 1 had the best PFS and OS; in contrast, the worst PFS and OS were found in group 4 (low p16 expression + clinical N2–3). The comparison of PFS and OS among different groups is shown in Figure 3.

## 4. Discussion

HNSCC is an aggressive cancer and the third leading cause of mortality among men in Taiwan. Growing evidence has confirmed that HPV status is a prognostic factor for HNSCC, particularly oropharyngeal cancer [3,4,5]. In general, p16 staining has been shown to predict the presence of HPV infection, and diffuse p16 unclear staining is usually considered HPV-positive [25]. However, the role of p16 cytoplasmic staining under negative p16 nuclear expression remains unclear. Our study enrolled 195 HNSCC patients who received TPF induction chemotherapy and showed that high expression of p16 cytoplasmic staining was a more favorable predictive factor for better response to induction chemotherapy than low expression. Moreover, patients with high expression of p16 cytoplasmic staining had superior PFS and OS compared with those with low expression, regardless of tumor size, tumor stage, or tumor location. The results of our study demonstrated the prognostic role of p16 cytoplasmic staining in patients with HNSCC. 

HPV has been proven to be associated with HNSCC, particularly oropharynx. A large systematic review of 39 studies with available clinical outcomes showed high heterogeneity in the diagnosis of HPV and the definition of p16. The correlation between positive HPV and overexpression of p16 was best in the group with ≥ 70% staining of p16 expression [25]. In general, diffuse p16 nuclear and cytoplasmic staining correlates with the presence of HPV infection. However, the status of p16 cytoplasmic staining under negative presentation of p16 nuclear expression is unclear; by definition, this status should be considered as HPV−. Our study showed high expression of p16 cytoplasmic staining to be predictive of a better response to induction chemotherapy and was an independent prognostic factor for better PFS and OS when the status was HPV- HNSCC.

In contrast, in our study, the percentage of high expression of p16 cytoplasmic staining in the oropharynx was 47%, which was not different from the percentage in the oral cavity (45%) and hypopharynx/larynx (41%). However, the oral cavity was an independent predictive factor of a worse response rate (52%) to TPF induction chemotherapy, but RR was up to 68% and 64% in the oropharynx and hypopharynx/larynx, respectively. This may be related to the clinical LN metastasis status: a borderline risk factor for poor response to induction chemotherapy in our study. In the analysis of survival, although the oral cavity as tumor location was a risk factor for poor response to induction chemotherapy, the 5-year PFS rate was 29%, higher than the 23% for tumors in the oropharynx, and the 5-year OS rate was higher for tumors in the oral cavity (32%) than those in the oropharynx (27%). As mentioned above, the percentage of high p16 cytoplasmic expression could not explain these outcomes, and the difference in PFS and OS may have been related to salvage surgery. In general, salvage surgery after CRT with residual tumor or recurrence is more applicable for tumors located in the oral cavity than in the oropharynx in clinical practice. However, salvage surgery could not explain the differences in PFS and OS between the oropharynx and hypopharynx. In general, tumors arising from the oropharynx and hypopharynx/larynx are suitable for organ/function preservation therapy, and the criteria for these two organs are similar. Our data revealed that patients with tumors located in the hypopharynx/larynx had better 5-year PFS rate (41% vs. 23%) and 5-year OS rate (47% vs. 27%) than those with tumors arising from the oropharynx. This may have been related to the status of LN metastasis, as the incidence of clinical N2–3 was higher in the oropharynx group (75%) than in the hypopharynx/larynx group (69%); clinical N2–3 was an independent prognostic factor for worse PFS and OS. In addition, patients with OS for more than 9 years (long-term survival) were also analyzed. Among them, all nine patients received concurrent CRT after induction chemotherapy TPF; six patients received CR after concurrent CRT, two patients underwent bilateral neck LN dissection due to residual tumor after concurrent CRT, and the remainder (one patient) experienced recurrence after concurrent CRT and salvage surgical resection were performed.

Our study had several limitations. First, there may have been some bias due to the retrospective design of our study. Second, there was a lower percentage of women patients ( < 10%); this may be due to the higher percentage of smoking, alcohol consumption, and betel nut chewing in our study, which are more common in men than in women. However, to the best of our knowledge, the current study provides limited evidence to investigate the role of p16 cytoplasmic staining in patients with HNSCC who received TPF induction chemotherapy; our findings may demonstrate the predictive and prognostic role of p16 cytoplasmic staining in these patients in clinical practice.

## 5. Conclusions

The results of our study showed that p16 cytoplasmic staining could predict treatment response to TPF induction chemotherapy and is an independent prognostic factor for PFS and OS in HNSCC. 

## Figures and Tables

**Figure 1 biomedicines-11-00339-f001:**
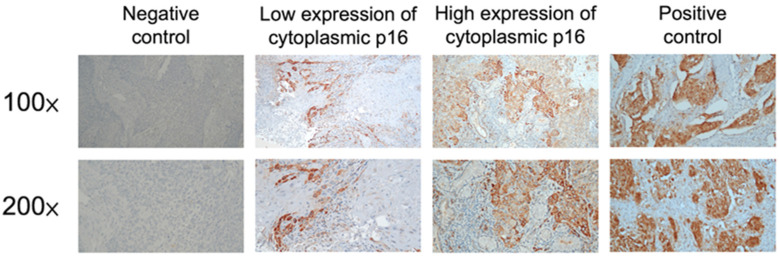
The immunohistochemical analysis of p16 in head and neck squamous cell carcinoma patients.

**Figure 2 biomedicines-11-00339-f002:**
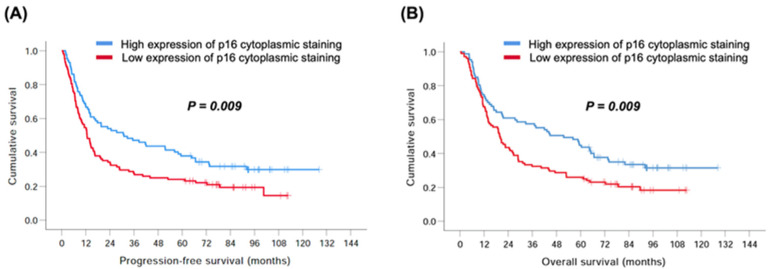
Comparison of Kaplan–Meier curves of progression-free survival and overall survival in 195 patients with head and neck squamous cell carcinoma based on the expression of p16 cytoplasmic staining. (**A**) Progression-free survival and (**B**) overall survival.

**Figure 3 biomedicines-11-00339-f003:**
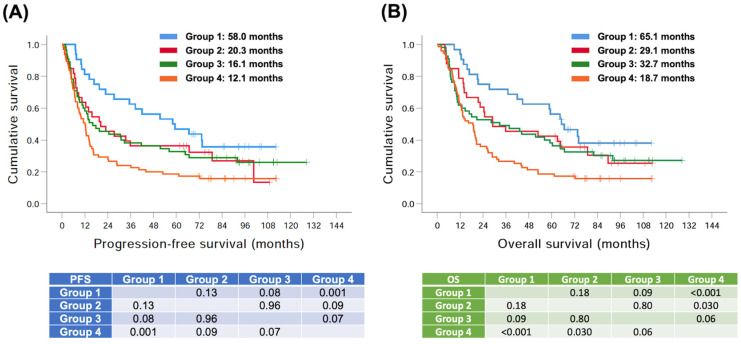
Comparison of Kaplan–Meier curves of progression-free survival and overall survival in 195 patients with head and neck squamous cell carcinoma based on the expression of p16 cytoplasmic staining and clinical N stage. (**A**) progression-free survival and (**B**) overall survival. Group 1: high p16 expression + clinical N0–1, *n* = 32, group 2: low p16 expression + clinical N0–1, *n* = 33; group 3: high p16 expression + clinical N2–3, *n* = 55; group 4: low p16 expression + clinical N2–3, *n* = 75.

**Table 1 biomedicines-11-00339-t001:** Characteristics of 195 patients with head and neck squamous cell carcinoma receiving TPF induction chemotherapy.

Age (Years)		
	Median (range)	52 (32–82)
Sex		
	Male	186 (95%)
	Female	9 (5%)
Primary tumor site		
	Hypopharynx	30 (15%)
	Larynx	21 (11%)
	Oropharynx	60 (31%)
	Oral cavity	84 (43%)
Tumor status		
	T1	5 (3%)
	T2	21 (11%)
	T3	27 (14%)
	T4a	64 (33%)
	T4b	78 (39%)
Nodal status		
	N0	41 (21%)
	N1	24 (12%)
	N2a	5 (3%)
	N2b	57 (29%)
	N2c	47 (24%)
	N3	21 (11%)
AJCC stage		
	III	15 (8%)
	IVA	83 (43%)
	IVB	97 (49%)
p16 cytoplasmic staining expression		
	Low	108 (55%)
	High	87 (45%)
Smoking		
	Absent	13 (7%)
	Present	182 (93%)
Alcohol		
	Absent	28 (14%)
	Present	167 (86%)
Betel nut chewing		
	Absent	40 (21%)
	Present	155 (79%)
Response to TPF induction chemotherapy		
	Complete response	10 (5%)
	Partial response	108 (55%)
	Stable disease	53 (27%)
	Progressive disease	24 (12%)

TPF: docetaxel, cisplatin, and fluorouracil; AJCC: American Joint Committee on Cancer.

**Table 2 biomedicines-11-00339-t002:** Associations between the p16 immunohistochemistry cytoplasmic staining and clinicopathologic parameters.

Parameters		p16 Cytoplasmic Staining Expression		
		Low	High	*p* Value
Age (years)	<52	50	45	0.45
	≧52	58	42	
Sex	Male	103	83	0.99
	Female	5	4	
Clinical 8th AJCC stage	III + IVA	52	46	0.51
	IVB	56	41	
Clinical T stage	T1–T4a	66	52	0.85
	T4b	42	35	
Clinical N stage	N0–1	33	32	0.36
	N2–3	75	55	
Primary tumor site	Oral cavity	46	38	0.88
	Others	62	49	
Primary tumor site	Oropharynx	32	28	0.70
	Others	76	59	
Primary tumor site	Hypo/Larynx	30	21	0.57
	Others	78	66	
Smoking history	Absent	6	7	0.49
	Present	102	80	
Alcohol history	Absent	16	12	0.84
	Present	92	75	
Betel nut chewing	Absent	17	23	0.07
	Present	91	64	

TPF: docetaxel, cisplatin, and fluorouracil. Chi-square test was used for statistical analysis.

**Table 3 biomedicines-11-00339-t003:** Associations between the response of TPF induction chemotherapy and clinicopathologic parameters.

Parameters		Response to Induction Chemotherapy		
		CR/PR	SD/PD	*p* Value
Age (years)	<52	59	36	0.66
	≧52	59	41	
p16 cytoplasmic staining	Low	57	51	0.014 *
	High	61	26	
Sex	Male	113	43	0.76
	Female	5	4	
Clinical 8th AJCC stage	III + IVA	62	36	0.43
	IVB	52	41	
Clinical T stage	T1–T4a	73	45	0.63
	T4b	45	32	
Clinical N stage	N0–1	45	20	0.08
	N2–3	73	57	
Primary tumor site	Oral cavity	44	40	0.043 *
	Others	74	37	
Primary tumor site	Oropharynx	41	19	0.14
	Others	77	58	
Primary tumor site	Hypo/Larynx	33	18	0.48
	Others	85	59	
Smoking history	Absent	8	5	0.94
	Present	110	72	
Alcohol history	Absent	17	11	0.98
	Present	101	66	
Betel nut chewing	Absent	26	14	0.52
	Present	92	63	

TPF: docetaxel, cisplatin, and fluorouracil; CR: complete response; PR: partial response; SD: stable disease; PD: progressive disease. * Statistically significant.

**Table 4 biomedicines-11-00339-t004:** Logistic models for the response of TPF induction chemotherapy.

Factors	Response of Induction Chemotherapy	
	OR (95% CI)	*p* Value
Low vs. high expression of p16 cytoplasmic staining	2.12 (1.15–3.89)	0.016 *
Primary tumor site: oral cavity versus others	2.07 (1.13–3.82)	0.019 *

TPF: docetaxel, cisplatin, and fluorouracil; OR: odds ratio; CI: confidence interval. * Statistically significant.

**Table 5 biomedicines-11-00339-t005:** Results of univariate log-rank analysis of prognostic factors for progression-free survival and overall survival.

Factors	Number of Patients	Progression-Free Survival (PFS)		Overall Survival (OS)	
		5-Year PFS (%)	*p* Value	5-Year OS (%)	*p* Value
Age (years)					
<52	95	25%	0.21	31%	0.33
≧52	100	35%		38%	
p16 cytoplasmic staining					
Low expression	108	24%	0.009 *	26%	0.009 *
High expression	87	38%		45%	
Clinical 8th AJCC stage					
III + IVA	98	34%	0.08	37%	0.09
IVB	97	27%		32%	
Clinical T stage					
T1–4a	118	32%	0.22	36%	0.25
T4b	77	27%		33%	
Clinical N stage					
N0–1	65	42%	0.009 *	49%	0.004 *
N2–3	130	25%		27%	
Primary tumor site					
Oral cavity	84	29%	0.21	32%	0.25
Others	111	32%		36%	
Primary tumor site					
Oropharynx	60	23%	0.49	27%	0.44
Others	135	33%		38%	
Primary tumor site					
Hypopharynx/larynx	51	41%	0.045 *	47%	0.046 *
Others	144	26%		30%	
Smoking history					
Absent	13	39%	0.36	46%	0.31
Present	182	30%		34%	
Alcohol history					
Absent	28	43%	0.21	46%	0.21
Present	167	28%		32%	
Betel nut chewing					
Absent	40	38%	0.30	43%	0.40
Present	155	28%		32%	

TPF: docetaxel, cisplatin, and fluorouracil. * Statistically significant.

**Table 6 biomedicines-11-00339-t006:** Results of multivariate Cox regression analysis for progression-free survival and overall survival.

Factor	Progression-Free Survival		Overall Survival	
	OR (95% CI)	*p* Value	OR (95% CI)	*p* Value
Clinical N stage: N2–3	1.64 (1.14–2.36)	0.007 *	1.75 (1.21–2.53)	0.003 *
Low expression of p16 cytoplasmic staining	1.47 (1.05–2.05)	0.024 *	1.49 (1.06–2.08)	0.021 *

* Statistically significant.

## Data Availability

The datasets used and analyzed during the current study are available from the corresponding author upon reasonable request.

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
