# Peer review of "Low p16 Cytoplasmic Staining Predicts Poor Treatment Outcome in Patients with p16-Negative Locally Advanced Head and Neck Squamous Cell Carcinoma Receiving TPF Induction Chemotherapy"

_biomedicines, 2023, doi:10.3390/biomedicines11020339_

Round 1
Reviewer 1 Report
1. The data given in table 1 are completely duplicated in the text of lines 151-174.
2. In table 5 network typos, please check.
3. The authors did not try to see how the overall survival rate changes when taking into account the factors from Table 6 at the same time? For example, Clinical N stage: N2–3 + Low expression of p16 cytoplasmic staining versus Clinical N stage: N0–1 + High expression of p16 cytoplasmic staining. For how many patients do both factors coincide?
Reviewer 2 Report
There are critical concerns. The following parts must be addressed and supplemented.
1. In the theoretical background, the basis for using P16 as a diagnostic marker for HNSCC should be described.
2. P16 is already a well-known prognostic marker of HNSCC. The relationship between low expression of P16 in HPV-negative HNSCC patients and poor prognosis is the same as in HPV-positive HNSCC patients.
3. According to Table 3, it appears that the percent of ORR is almost the same regardless of whether P16 is expressed or not. If so, it is very likely that the anticancer drugs (TPF) did not affect the therapeutic effect actually and that other care was involved in the relationship between long-term survival and P16 expression. In addition to the showing survival curve, analysis for all treatments during long-term survival (about 108-150 months) should be accompanied.
4. According to Tables 4 and 5, there are independent variables such as N stage and primary tumor site, and they are statistically significant differences (although there is no significance, AJCC stage can be considered as well). This is determined to be an error in the patient's group settings. If there are other variable factors, it is unreliable to define the difference in survival only by the expression condition of P16.
Reviewer 3 Report
This manuscript, written by Dr. Chen, original research, with the title of "Low p16 cytoplasmic staining predicts poor treatment outcome in patients with HPV-negative locally advanced head and neck squamous cell carcinoma receiving TPF induction chemotherapy" correlated the immunohistochemical expression of p16 with the prognosis of patients of head and neck squamous cell carcinoma. This research is different from other because it focuses only in the cytoplasmic staining, and not in the nuclear. The authors found that high cytoplasmic expression correlated with a favorable prognosis of the patients. The manuscript is well written, and it is easy the read and to understand. There are enough figures and tables.
To improve the manuscript, a series of comments/questions could be addressed:
1) The gold standard for assessing HPV infection in patients with HPV associated head and neck cancer is in situ hybridization or PCR to detect HPV DNA. Do you have data using the gold standard method? Does the gold standard correlated with the immunohistochemistry?
2) Immunohistochemistry (IHC) for p16 is highly sensitive for HPV associated tumors, and we routinely use it as a surrogate for HPV status. Usually, strong and diffuse, nuclear and cytoplasmic staining by p16 (at least "70%" of the tumor cells) correlates with the HPV infection. In lines 118-119, it is stated that a cut-off of nuclear 50% was used to classify HPV- from +. Is this cut-off the "official" threshold of positivity?
3) Could you please describe in a paragraph the pathogenetic pathway of HPV in oropharyngeal squamous cell carcinoma, including HPV E6 and E7 mRNA, p53, RB proteins, and indirect accumulation of p16? A figure could be useful as well.
4) Line 121, it is stated that this series of 195 cases was "p16-negative" and "HPV-negative". Should this be included in the abstract? The title uses HPV-negative, but without PCR... Is it right? Did nuclear and cytoplasmic staining correlated?
5) Regarding Figure 1. What are the negative and positive controls?
6) Could you please show Figure 1 at higher definition? For each figure, could you please write the intensity and percentage of cells?
7) Regarding Table 4. Is it a binary logistic regression? Which is the reference value? Low p16 expression correlated with response to therapy, but worse survival?
8) Line 258. "Our study showed high expression of p16 cytoplasmic staining to be predictive of better response to induction chemotherapy and was an independent prognostic factor for better PFS and OS, when the status was HPV+ HNSCC." Sorry, but, were not HPV-negative?
9) Did you analyze cases that were HPV-positive? Are they in another publication?
10) Do you data correlates well with other similar publications? Other publications use nuclear/cytoplasmic diffuse expression....
11) What is the biology of p16 cytoplasmic expression? Does it reflect a different pathogenesis?
12) Could you please comment on high risk and low risk HPV subtypes?
13) Has this antibody been used by other groups? Is it as sensitive for HPV as other already published primary antibodies?
Round 2
Reviewer 2 Report
All concerns have been well addressed.